# Non-Conserved Amino Acid Residues Modulate the Thermodynamics of Zn(II) Binding to Classical ββα Zinc Finger Domains

**DOI:** 10.3390/ijms232314602

**Published:** 2022-11-23

**Authors:** Katarzyna Kluska, Aleksandra Chorążewska, Manuel David Peris-Díaz, Justyna Adamczyk, Artur Krężel

**Affiliations:** Department of Chemical Biology, Faculty of Biotechnology, University of Wrocław, Joliot-Curie 14a, 50-383 Wrocław, Poland

**Keywords:** metal binding affinity, energetics, DNA, isothermal titration calorimetry, metal-coupled folding, molecular dynamics, hydrogen bond

## Abstract

Classical zinc fingers domains (ZFs) bind Zn(II) ion by a pair of cysteine and histidine residues to adopt a characteristic and stable ββα fold containing a small hydrophobic core. As a component of transcription factors, they recognize specific DNA sequences to transcript particular genes. The loss of Zn(II) disrupts the unique structure and function of the whole protein. It has been shown that the saturation of ZFs under cellular conditions is strictly related to their affinity for Zn(II). High affinity warrants their constant saturation, while medium affinity results in their transient structurization depending on cellular zinc availability. Therefore, there must be factors hidden in the sequence and structure of ZFs that impact Zn(II)-to-protein affinities to control their function. Using molecular dynamics simulations and experimental spectroscopic and calorimetric approaches, we showed that particular non-conserved residues derived from ZF sequences impact hydrogen bond formation. Our in silico and in vitro studies show that non-conserved residues can alter metal-coupled folding mechanisms and overall ZF stability. Furthermore, we show that Zn(II) binding to ZFs can also be entropically driven. This preference does not correlate either with Zn(II) binding site or with the extent of the secondary structure but is strictly related to a reservoir of interactions within the second coordination shell, which may loosen or tighten up the structure. Our findings shed new light on how the functionality of ZFs is modulated by non-coordinating residues diversity under cellular conditions. Moreover, they can be helpful for systematic backbone alteration of native ZF ββα scaffold to create artificial foldamers and proteins with improved stability.

## 1. Introduction

Common zinc finger domains (ZFs) coordinate Zn(II) ions by a pair of cysteine and histidine residues (Cys_2_His_2_ or CCHH) to adopt a stable ββα fold with an inner hydrophobic core (Figure 1A). Such a unique protein fold offers an attractive DNA recognition code, so it is not surprising that ZFs are widespread in nature, from archaea to eukaryotes. Thus, ZFs are widely applied for various applications in science. The systematic backbone alteration of the native ZF backbone, where unnatural building blocks are integrated, is a promising strategy to generate heterogenous-backbone foldamer with improved stability [1,2]. Furthermore, the ZF scaffold can be conjugated to a photo-responsive azobenzene unit, allowing for a tertiary structure formation and hence providing a powerful basis for application in areas such as photodynamic gene therapy and synthetic biology [3,4]. Moreover, due to ZFs’ flexibility and ability to be modified to recognize diverse DNA sequences, ZFs have emerged as a powerful tool for gene editing [5]. Recent progress in developing systems that may impact ZF-DNA interaction is constantly growing, and new chimeric proteins are being described. By fusing zinc finger peptide to repression or activation domains, genes can be selectively switched off or on [6,7,8]. The design of chimeric ZFs exploited by combining two ZF backbones into one single chain enables protein mimics with defined tertiary structures. Such a redesign can be seen as a promising approach to creating artificial proteins with properties and applications beyond those known for natural motifs. However, changing an entire structural element such as an α-helix or amino acid composition without causing major unfolding remains a challenge and requires broad knowledge in terms of structure and side-chain distributions. Therefore, to design or even mimic behavior of ZFs, it is important to understand how sequential composition can alter the structure–stability–function relationship.

Collectively, a growing body of evidence suggests that the loss of Zn(II) immediately causes disruption of the unique structure and function of the whole domain [9,10]. Too low affinity of ZFs to Zn(II) would not guarantee saturation under cellular conditions (when the *K*_d_ value is higher than free Zn(II) concentration), or ZFs saturation can occur transiently upon Zn(II) fluctuations. It has been shown that numerous ZF-containing proteins are saturated under specific cell conditions [9,10,11,12,13]. Therefore, there must be factors hidden in the sequence and structure of ZFs that impact Zn(II)-to-protein affinities to control ZFs function. Some of them are strictly related to conserved sequences, such as the composition of metal binding residues and the presence of a hydrophobic core. For example, eliminating one of the metal binding residues in ββα ZFs significantly decreases metal ion affinity and disturbs ligand geometry [9,14,15]. Nevertheless, a significant difference in Zn(II) affinity is still observed in many non-truncated classical ZFs with high sequence conservation, raising the question of whether or not non-conserved residues can alter ZF stability. If so, what are the factors that govern such stability adjustments? Understanding the role of particular non-conserved residues opens a new avenue for protein engineering and exploration of interaction mechanisms of ZFs with biomolecules at the molecular level. Therefore, in this report, we aimed to investigate how sequential variations present in consensus peptide 1 (CP1) sequences, defined in 1991 based on only 131 ZF sequences [16,17] and later redefined in 2015 based on 13,456 ZF sequences [18] (Figure 1), impact the thermodynamics of ZFs. Using molecular dynamics simulations, we showed that sequence variation around non-conserved amino acids influence the pathway of metal-coupled folding mechanism and hydrogen bonds (h-bonds) formation. This was further proved by experimental spectroscopic studies and thermodynamics ITC measurements. Such insightful data regarding the scenario of ZF sequence evolution from CP1-1991 to CP1-2015 enabled us to show how sequential composition impacts heterogeneity throughout ZFs stability and describe which non-conserved residues are important in stability alteration without causing major loss of ββα ZF structure.

**Figure 1 ijms-23-14602-f001:**
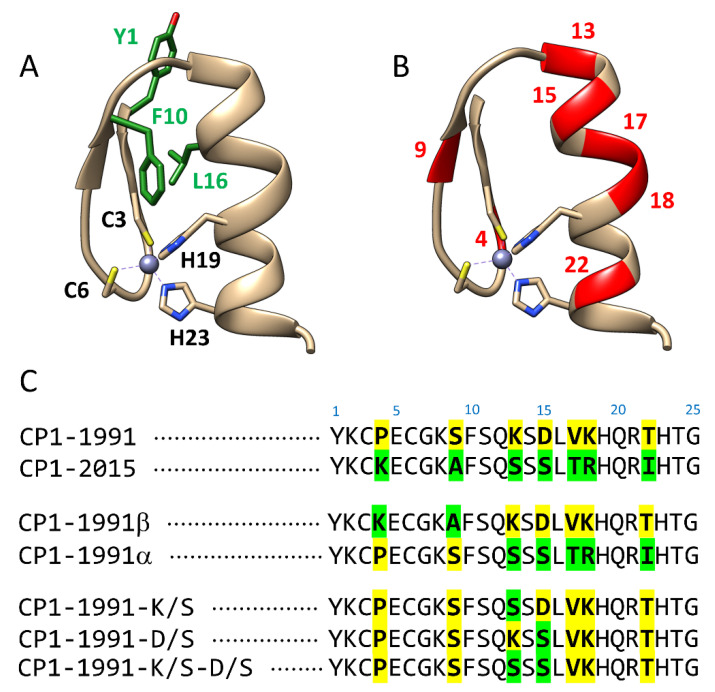
Structure and sequences of zinc finger peptides obtained and examined in this work. (**A**) Structural representation of consensus zinc finger peptide CP1 with an indication of Zn(II) binding (C3, C6, H19, and H23) and hydrophobic residues (green). Figure prepared using Chimera v. 1.13.1 based on the structure of the second ZF of the designed zinc finger protein (PDB: 1MEY) [17]. (**B**) Variable residues present in both CP peptides are marked in red. (**C**) Sequences of ZF peptides used in this study. Yellow and green colours represent amino acid residues of CP1-1991 and CP1-2015, respectively, located at variable positions.

## 2. Results and Discussion

Although in this study, the greatest emphasis is placed on the understanding of energetic consequences of the sequential difference in CP1-1991 and CP1-2015 zinc fingers, single and double mutants were used in addition to these peptides (Figure 1B). They have been used to investigate the influence of particular ZF regions and residues on Zn(II) binding thermodynamics.

### 2.1. Classical Molecular Dynamics Simulations

The dynamic effects between the CP1 peptides were first investigated by classical MD simulations (Materials and Methods). To check whether there are significant differences in the overall conformational dynamics, the root-mean-square deviation (RMSD) was monitored as a function of the time with respect to their initial conformation. As shown in Appendix A, the RMSD of both peptides exhibits similar characteristics, indicating that the non-conserved amino acid residues have a minor effect on the global conformation. In all of the simulations, Zn(II) remained bound to the protein, and subsequently, we did not observe any metal-coupled folding process characteristic of zinc fingers [19]. We then studied the individual residue flexibility by calculating the root-mean-square fluctuations (RMSF) of the α-carbons for each system and trajectory independently. Overall, the plot shows that the N-terminus is relatively more disordered and flexible than the C-terminus α-helix, and no significant differences were observed (Appendix A). We then attempted to characterize the conformational space sampled during the classical MD simulations by performing a principal component analysis (PCA) on the α-carbons and estimating the free energy surface (FES) as a function of the first and second principal components [20]. The estimated FES showed three minima energy in both systems, indicating once more that both systems sampled similar configurations (Appendix A). The principal component (PC) 1 was related to the flexibility in the C-terminus, whereas the N-terminus and the middle part of the protein contributed the most to the PC2 (Appendix A). However, no clear differences were observed between both CP1-1991 and CP1-2015. Afterward, structures were extracted from each minimum energy identified in the FES, and h-bonds and salt bridges were calculated to unveil their molecular features. Extracted representative structures indicated three h-bonds stabilizing the α-β helix-sheet interaction in the CP1-1991 but not in the CP1-2015 system (Figure 2). Moreover, we could observe a larger α-helix content that is translated in one h-bond more in the CP1-1991, which gives a rational explanation for the higher α-helix stability (Figure 3A). One would expect the formation of h-bonds closer to the C-terminal part, as it is where it lies differences in the α-helix between both peptides. The distant location of the three h-bonds from the C-terminal α-helix suggests an allosteric role in these intermolecular interactions. Focusing on the Zn(II)-binding site, some structural differences also appeared, which resulted in the displacement of the His23; thus, it shifted the Zn-N bond distance (Figure 3B).

### 2.2. Steered Molecular Dynamics Simulations

To further explore the structure–stability relationship, we investigated Zn(II) (un)binding mechanism to ZFs by non-equilibrium steered MD simulations (SMD). In this work, the free energies derived from SMD simulations cannot usually be quantitatively compared with experimental data because of the use of fast-pulling regimes and the improper electronic description of the Zn(II) site that would require quantum chemical treatment [21,22]. However, SMD has been shown to be a particularly relevant method for identifying molecular mechanisms [23,24,25]. Moreover, when the proper variables that may affect the results have been carefully considered (e.g., speed and pulling force), potential mean forces can be compared between different systems [26]. Forty independent constant-speed SMD simulations for each ZFs were performed to obtain a statistical distribution of Zn-donor unbinding events.

The process was studied by pulling apart the Zn(II) while the protein remained fixed. Three possible mechanisms were identified for CP1-2015 that shared common intermediates: ZnL1, ZnL2, and ZnL3 (Appendix A). In all of the mechanisms, Cys3 was the first residue that dissociated from Zn(II), and Cys6 was the last residue coordinating Zn(II) (Appendix A). The mechanisms differed in the ZnL2 intermediate, indicating that either both His are bound with similar strength or that the fast-pulling regime used in the SMD impeded their differentiation. In one of the pathways, Cys6 and His19 bind Zn(II) in ZnL2, and another pathway, Cys6 and His23 bind Zn(II) in ZnL2 (Figure 4). In the case of CP1-1991, one dominant pathway (87% repeated) was found to be similar to one pathway in CP1-2015. The mean rupture force, which indicates the mean force required to break all of the Zn-L bonds, was larger for CP1-1991 than for any pathway in CP1-2015 (Appendix A). The total work done followed similar trends to the mean rupture force (Appendix A). To further strengthen our results, we performed SMD by using CP1-1991α mutant (Figure 1). Similar to CP1-1991, one dominant pathway (85% occurrence) with identical stepwise Zn-L bond dissociation was found (Figure 4 and Appendix A). The simulations also pointed out that the first Zn-L dissociation where L corresponds to Cys3 is the event that requires the higher force or work and that both His are bound with low and similar strength (Appendix A).

### 2.3. Spectroscopic Characterization

Our MD simulations clearly show that CP1-1991 is more stable than CP1-2015 due to an additional stabilizing effect derived from h-bond formation between non-conserved residues found in C-terminus α-helix and conserved hydrophobic residues (Figure 2), as well as additional h-bond stabilizing α-helix (Figure 3). The mean rupture forces inform the strength of Zn-L interactions; however, this value reflects the mean for the four Zn-L interactions, but does exclude how the protein affects the Zn(II) unbinding? It is measured by the total work done. The peptide CP1-1991 yields the highest total work done, and the α-helix replacement (CP1-1991α) causes a decrease to a value close to that obtained for CP1-2015. In the next step, we aimed to experimentally investigate by spectroscopic Zn(II) and Co(II) titration and isothermal titration calorimetry (ITC) whether or not non-conserved amino acids can be responsible for stability alteration without inducing major structural changes.

A major focus was placed on CP1-1991 and CP1-2015, then residues from the β-fragment of CP1-1991 were substituted with those from CP1-2015 (CP1-1991-β) and the same for alfa fragment (CP1-1991α) to probe which structural element from CP1-2015 are responsible for its stability loss in comparison to CP1-1991 (Figure 1). Finally, peptides with single and double mutations were also synthesized. In this case, non-conserved residues from the CP1-1991 α fragment, indicated by MD simulations as prone to form h-bond (Lys13 and Asp15) (Figure 2), were replaced step-by-step to those from the CP1-2015 α fragment (CP1-K/S, CP1-D/S, CP1-K/S-D/S) to probe if loss of h-bonding impact free energy of Zn(ZF complex formation (Figure 1). Following that, the CD spectra of all ZFs were recorded to examine structural changes during Zn(II) coordination.

All CD spectra feature similar band patterns with two negative bands at 210 and 220 nm, indicating the presence of functional ββα fold (Appendix A). In all cases, CD titrations showed 1:1 (ML) stoichiometry. Then UV-Vis-based Co(II) titrations were performed to follow metal binding mode and the coordination of metal ions in complex species. Absorbance changes clearly demonstrate that all ZF peptides form CoL complexes with two visible d-d bands at ~650 and ~580 nm with high molar absorption coefficients greater than 300 M^−1^ cm^−1^ (Appendix A). The position and pattern of d-d bands confirm typical tetrahedral Zn(Cys_2_His_2_) coordination found in many ββα ZFs [9]. In the last step of the spectroscopic studies, the dissociation constant (*K*_d_) values of ZnZF complexes were determined by competition study with metal chelators (having various affinities for Zn(II)). Results show that amino acid residues from CP1-2015 β fragment do not participate in the significant stability loss, while amino acids residues from the alfa region of CP1-2015 are crucial in intrinsic stability loss from CP1-1991 to CP1-2015 (Appendix A, Table 1), which is in agreement with MD simulations.

However, it should be pointed out that even though all amino acid residues are from the CP1-2015 α-region, the CP1-1991α ZF is still able to maintain a more stable ZnZF complex than CP1-2015. It is probably attributed to the presence of Pro4 and Ser9 (from β-fragment of CP1-1991). In general, the proline residue has been shown to stabilize the local three-dimensional structure of proteins by reinforcing hydrophobic interactions and by reducing the flexibility of a loop [26,27,28], while polar side chains of serine have a strong tendency to form h-bond between side chain and neighboring backbone amides [29,30]. This is also in agreement with the observed slightly lower affinity for CP1-1991β in comparison to CP1-1991, however, in this case, the difference in destabilizing effects induced by the loss of Pro4 and Ser4 residues is compensated by stabilizing h-bond interactions found in CP1-1991 α-region (Figure 2).

### 2.4. Isothermal Titration Calorimetry

To further understand the relative thermodynamic contributions of h-bond interactions found in CP1-1991 ZF peptide to the overall stability, insightful thermodynamic analysis was performed (Figure 5). In this case, the Zn(II) binding to ZF peptides such as CP1-1991-α, -K/S, -D/S, and -K/S-D/S were examined using isothermal titration calorimetry. Table 2 represents obtained thermodynamic parameters (∆*G*°, ∆*H*°, −T∆*S*°, ∆*H*_Zn-pep_, ∆*H*_CysH_, and ∆*H*_folding_) (Material and Methods) [31]. The number of protons dissociated upon Zn(II) binding (*n*_H_) was calculated based on acid dissociation constants of cysteine thiols derived from pH-metric titrations of ZF peptides (Appendix A) [32,33]. The enthalpic and entropic contribution of Zn(II) binding to CP1-1991-β are very similar to CP1-1991, however, the overall binding enthalpy (∆*H*°) is about 0.44 kcal/mol less favorable for CP1-1991-β. This enthalpic loss in total binding enthalpy is predominantly due to enthalpic contribution from thiols deprotonation, as the values of ∆*H*_Zn-pep_ for these two peptides are almost identical, showing that there are no significant differences derived from protein folding to the binding enthalpy among these peptides [33]. On the other side, in comparison to CP1-1991, the CP1-2015 and CP1-1991-α peptides exhibit similar entropic contributions and 2.48 kcal/mol and 1.96 kcal/mol less favorable overall enthalpy, respectively. Furthermore, similarly to CP1-1991, the Zn(II) binding to CP1-2015 and CP1-1991-α is enthalpically driven; however, for the other ZF peptides, the overall enthalpy changes (∆*H*°) become less favorable while the entropy of binding becomes more favorable for the CP1-1991-α, -D/S, and -K/S/-D/S, to the point where Zn(II) binding for the CP1-K/S is entropically driven (Table 2, Figure 6). Such an entropy advantage has been previously observed mostly for CCCC and CCHC ZFs, where a negative increase in entropic component was consistent with an increasing number of dissociated protons [31,33,34]. Nevertheless, this is not the case for the CP1-1991-α, -D/S, K/S, and -K/S/-D/S ZFs, as they displace an almost identical number of protons (Table 2), suggesting that entropy of binding depends on other factors. Even though this increase in an entropic component is harder to estimate, it can be seen that the value of −TΔS°, for these peptides, becomes increasingly negative as enthalpy of protein folding (∆*H*_folding_) becomes less favorable. It suggests that non-conserved amino acid residues such as Lys13 and Asp15 are involved in h-bond interaction with conserved aromatic residues, as indicated by MD simulation. However, it should be pointed out that the most significant decrease in Δ*H*° values was observed for the CP1-1991-K/S peptide. The ∆*H*° values are similar and become more favorable for the CP1-1991-D/S and -K/S-D/S peptides, suggesting that the substitution of Asp15 compensates unfavorable entropic gain observed for the CP1-1991-K/S. It proves that heterogeneity among non-conserved residues can adjust ZF stability by a pool of available backbone interactions. Therefore, loss in these interactions may affect hydrophobic-core packing, leading to a crevice at the protein surface and fluctuations in the positions of surrounding side chains, which leads to entropic gain.

### 2.5. Comparison with Previous Reports and Biological Significance

The thermodynamic data derived in our study can be directly related to the sixth CCHH ZF of the human male-associated ZFY protein [35,36]. In particular, an entropic gain associated with the hydrophobic core packing and increased degree of freedom caused by loss of non-covalent interactions has also been reported. However, in the case of the ZFY protein, these entropic changes have been measured for a specific aromatic residue [34,35,36]. Here, we show that loss of non-conserved variable amino acid residue located at a particular position can also lead to entropic gain and modulate Zn(II)-binding thermodynamics. As indicated by both computational and experimental results, different enthalpy and entropy contributions were related to the gain or loss of specific h-bond interactions (Figure 2), as well as to structural differences in Zn(II) binding site between CP1-1991 and CP1-2015 (Figure 3). Overall, we found a good agreement between the computational and experimental results, which allowed us to propose an atomistic view of the general Zn(II) unbinding mechanism (Appendix A). Moreover, key non-conserved residues that form specific h-bond interactions enhancing the thermodynamic stability of the CP1-1991 ZF were discovered. These results show that adequately applied backbone mutations in CCHH ZF allow for Zn(II) binding affinity adjustment with subsequent preservation of the ββα fold.

It is also worth underlying that the CP1-2015 consensus peptide is a better thermodynamic model of classical ZFs compared to CP1-1991 due to its moderate affinity for Zn(II). This affinity (−log*K*_d_^7.4^ = 12.30) more adequately represents the affinity of average ZF domains investigated over the last three decades [9]. The unusually high affinity of CP1-1991 found here as low femtomolar (−log*K*_d_^7.4^ = 14.49), or even higher according to others [11], is due to the abovementioned stabilizing interactions. These interactions likely occurred more often in ZF sequences (131 only) identified in early studies. A new consensus sequence, based on thousands of ZF sequences, represents the average status of stabilizing interactions present in classical ZFs. Therefore, if one is interested in a more biological model in terms of ZF thermodynamics, the new CP1-2015 is a better choice. However, old CP1-1991 is likely a more suited peptide model if a very high affinity is needed.

Our results also shed light on how proteins containing various ZFs behave under cellular conditions. Zinc finger motifs composing more distinct intramolecular stabilizing interactions bind Zn(II) with high affinities. Their dissociation constants are below intracellular free Zn(II) concentrations, which consequently warrant their Zn(II) saturation and structural functionality. Depending on cell types, the concentration of freely available Zn(II) in eukaryotes varies from the low nanomolar to the picomolar range [13,37,38] and remains a few orders of magnitude higher than *K*_d_ values of the tightest ZFs. ZF motifs, containing less or even lacking in stabilizing effects, bind Zn(II) with much lower affinities, which has been demonstrated on several occasions [9,11,12,14]. The affinity is strictly related to the ZF sequence and quality of the determination, but it varies from the nanomolar to the picomolar range of *K*_d_ values. This range matches fluctuations of free Zn(II) concentrations occurring under various stimuli. Comparable *K*_d_ and free Zn(II) concentrations indicate that these ZF motifs might be transiently saturated when zinc availability is normal or increases under certain conditions. When cells are zinc deficient, structural Zn(II) ions dissociate from those sites, and ZF-containing proteins remain inactive. Zn(II) association and dissociation investigation have been performed recently on yeast cells, indicating that zinc metalloproteome is a highly dynamic system [13,39].

## 3. Materials and Methods

### 3.1. Materials

The following reagents were purchased from Sigma-Aldrich (Merck): nitrilodiaceticpropionic acid (NDAP), ethylene glycol-bis (2-aminoethylether)-*N*,*N*,*N*′,*N*′-tetraacetic acid (EGTA), *N*-(2-hydroxyethyl) ethylenediamine-*N*,*N*′,*N*′-triacetic acid (HEDTA), ethylenediamine-*N,N*′-disuccinic acid (EDDS), ethylenedinitrilotetraacetic acid (EDTA), *N*,*N*,*N*′,*N*′-tetrakis (2-pyridylmethyl) ethylenediamine (TPEN), 4-(2-hydroxyethyl)piperazine-1-ethanesulfonic acid (HEPES buffer), tris hydrochloride (Tris-HCl buffer), zinc(II) sulphate heptahydrate (ZnSO_4_·7H_2_O), cobalt(II) nitrate hexahydrate (Co(NO_3_)_2_·6H_2_O), 1,2-ethanedithiol (EDT), thioanisole, anisole, and triisopropylsilane (TIPS). The metal-chelating resin Chelex 100 was acquired from Bio-Rad. *N*,*N*-dimethylformamide (DMF) and 98% HCl were purchased from VWR Chemicals. Acetonitrile (ACN) was acquired from Merck Millipore. Sodium chloride (NaCl), sodium perchlorate (NaClO_4_), acetic anhydrine, diethyl ether, and dichloromethane (DCM) were purchased from Avantor Performance Materials Poland (Gliwice, Poland). Tris(2-carboxyethyl)phosphine hydrochloride (TCEP), 1-methyl-2-pyrrolidinone (NMP), *N*,*N*,*N*′,*N*′-tetramethyl-*O*-(1*H*-benzotriazol-1-yl)uronium hexafluorophosphate (HBTU), trifluoroacetic acid (TFA), *N*,*N*-diisopropylethylamine (DIEA), piperidine, TentaGel S Ram, and Fmoc-protected amino acids were obtained from Iris Biotech GmbH (Marktredwitz, Germany). Buffers’ pH was adjusted using either hydrochloric acid (NORMATOM, HCl) or ultra-pure sodium hydroxide purchased from VWR or Avantor Performance Materials Poland, respectively. The concentration of metal ion salt stock solutions was 0.05 M and was confirmed by a representative series of ICP-MS measurements. All pH buffers were treated with Chelex 100 resin to eliminate trace metal ion contamination.

### 3.2. Computational Studies

#### 3.2.1. Molecular Dynamics (MD) Simulations

The initial structure was obtained as in our previous report. Briefly, there is no solved structure for the ZF consensus, and thus homology modelling was employed by using the NMR structure PDB:2YTR. VMD was used to produce point mutations [40]. The MD simulations were performed with GROMACS 2018.4 [41]. The protonation states of the side chains at pH 7.0 were assigned using PROPKA, except for the Cys residues, which were deprotonated. The AMBER FF19SB force field and recently published cysteine/histidine-Zn(II) force field parameters were used to model the protein and the Cys and His residues [42]. The protein was solvated in an 8 Å cubic box filled with TIP3P water molecules and NaCl was added to achieve neutrality. A four-step protocol was used to equilibrate the systems. First, the steepest descent minimization (10,000 steps) was applied to the system and was followed by heating up the system from 0 to 300 K in the NVT ensemble, using the Langevin thermostat with a damping coefficient of 1 ps^−1^. In the third step, the system was equilibrated at constant pressure and temperature (NPT) for 100 ns at 1 atm and 300 K using Berendsen weak coupling. In the last step, 100 ns were run using the Parinello–Rahman barostat and the Nosé–Hoover thermostat. The Particle Mesh Ewald (PME) algorithm was used to evaluate electrostatic interactions using a cut-off of 8 Å. The LINCS algorithm was used to constraint bonds only involving hydrogen atoms to allow the use of a 2 fs time step. Finally, three independent production runs of 300 ns were obtained for each system. In the analysis of intermolecular interactions, an H bond was considered between an atom with a hydrogen bond and another atom when the distance between them was less than 3 Å and the donor–hydrogen–acceptor angle was less than 20 degrees. A principal component analysis (PCA) was performed on the α-carbons and the PCA eigenvalues were used to calculate free energy surfaces so that free energies were calculated from probabilities of occupying different states. The RMSD, RMSF, PCA, and free energy surfaces were calculated using the bio3d R package [43]. The figures were prepared using ggplot2 and UCSF Chimera [44,45].

#### 3.2.2. Steered Molecular Dynamics (SMD) Simulations

Constant-speed SMD simulations were used to study the Zn(II)-unbinding of the ZFs. Forty independent SMD runs at constant speed were performed for each ZF, using the distance between the Zn(II) and the center of mass of the initial four ligand residues where it was coordinated as reaction coordinates (RC). To avoid distortions in the protein backbone due to the force applied, 10 kcal·mol^−1^ positional restraints were applied to all of the CA atoms. Three spring constants (10, 250, 500 kcal·mol^−1^) and three velocities (0.1, 1, 10 Å·ns^−1^) were tested to determine the pulling regime that allowed us to discern between the Zn-L ligands and the different ZFs peptides. A force constant of 250 kcal·mol^−1^ and pulling speed of 1 Å·ns^−1^ were chosen, and as a result, each SMD pulling time was 7 ns, giving a total SMD simulation of 840 ns. The SMD simulations were performed with GROMACS 2018.4 in combination with the PLUMED 2.6 plugin [46]. To identify Zn-L dissociation pathways, the contact number (CN) between Zn(II) and each ligand was determined using the CN defined in Equation (1):(1)$${\mathrm{CN}}_{\mathrm{Zn}-S}=\underset{i\in A}{\sum }\underset{i\in B}{\sum }s_{ij}$$
where *A* is the Zn(II) ion, *B* corresponds to the ligand residue, and *s_ij_* is a switching function. The switching function is defined as presented in Equation (2):(2)$$s_{ij}=\frac{1-{(\frac{r_{ij}}{r_{0}})}^{n}}{1-{(\frac{r_{ij}-}{r_{0}})}^{m}}$$
where *n* = 8 and *m* = 12 and they define the steepness of the switching function, and *r*_0_ = 2.8 Å, which defines the cut-off to where the interactions between Zn(II) and the ligand atom are calculated.

### 3.3. Zinc Finger Peptide Synthesis

All investigated zinc finger peptides were synthesized via solid phase synthesis on TentaGel S Ram resin (0.22 mmol/g substitution) using the Fmoc- strategy and a Liberty 1 microwave-assisted synthesizer (CEM). Cleavage and purification were performed as previously described [47]. Acetic anhydrate was used for N-terminal acetylation, then peptides were cleaved from the resin with a mixture of TFA/anisole/thioanisole/EDT/TIPS (88/2/3/5/2, *v*/*v*/*v*/*v*/*v*) over a period of 2.5 h followed by precipitation in cold (−70 °C) diethyl ether. The crude peptide was collected by centrifugation, dried, and purified using HPLC (Waters 2487) on Phenomenex C18 columns using a gradient of ACN in 0.1% TFA/water from 0% to 40% over 20 min [48]. Purified peptides were identified by an API 2000 ESI-MS spectrometer (Applied Biosystems) or Compact Q-TOF ESI-MS (Bruker Daltonics, Bremen, Germany). The calculated (MW_cal_) and experimental (MW_exp_) molecular masses of synthesized zinc finger peptides are presented in Appendix A and they refer to averaged, not monoisotopic, values.

### 3.4. Determination of pK_a_ Values of ZF Thiols

In order to determine p*K*_a_ values of cysteine thiols (p*K*_a_^Cys^), spectrophotometric pH-titrations of metal-free ZF peptides were performed on a Jasco V-650 spectrophotometer (JASCO) at 25 °C in a 1 cm quartz cuvette over the UV range of 210–340 nm. For that purpose, 25 µM zinc finger peptide solutions were prepared in 0.1 M NaClO_4_, acidified to pH ~3, and then titrated with 0.1 M NaOH in a pH range from ~3 to ~8 under argon atmosphere. The absorption increase at 218 nm was taken for the calculation of the p*K*_a_^Cys^ of each peptide. The experimental data were fitted to Equation (3):(3)$$A=\frac{A0+A1\times 10^{\mathrm{pH}-\mathrm{pK}1}+A2\times 2\times 10^{2\times \mathrm{pH}-\mathrm{pK}1-\mathrm{pK}2}}{1+10^{\mathrm{pH}-\mathrm{pK}1}+2\times 10^{2\times \mathrm{pH}-\mathrm{pK}1-\mathrm{pK}2}}$$
where pK1 and pK2 are p*K*_a1_^CysH^ and p*K*_a2_^CysH^, respectively. A0, A1, and A2 correspond to experimental minimal, intermediate, and maximal absorbance values, respectively.

### 3.5. Circular Dichroism

Circular dichroism (CD) spectra of ZF peptides were recorded using a J-1500 Jasco spectropolarimeter (JASCO) at 25 °C in a 2 mm quartz cuvette, under a constant nitrogen flow over the range of 198–260 nm with a 100 nm/min speed scan. Final spectra were averaged from three independent scans. Spectroscopic titrations of 25 µM ZF peptide with Zn(II) ions were performed in chelexed 20 mM Tris-HCl buffer (100 mM NaCl, pH 7.4) with the addition of 10 mM TCEP (pH 7.4) to a final concentration of 200 μM as a non-metal binding cysteine thiol protector [49]. All samples were equilibrated over 2 min after the addition of each portion of 2 mM ZnSO_4_ solution. CD signals in mdeg were converted and analysed as molar ellipticity in deg·cm^2^·dmol^−1^ units.

### 3.6. Co(II) Binding to ZF Peptides

UV-Vis spectra were recorded using a Jasco V-650 spectrophotometer (JASCO) at 25 °C in a 1 cm quartz cuvette over the range of 350–800 nm. Spectroscopic titrations of 30 µM ZF peptides were performed in chelexed 50 mM HEPES buffer (100 mM NaCl, pH 7.4) with 2.5 mM Co(NO_3_)_2_ to a final concentration of 100 µM. The TCEP was added as a reducing agent to at least 4 eq. excess over each cysteine thiol. All samples were equilibrated for 2 min after the addition of each portion of Co(II) stock solution [14].

### 3.7. Zinc Finger Competition with Chelators

In order to determine Zn(II)-to-ZF affinity, peptides at 25 μM concentrations were equilibrated in a 1.0 mM solution of various chelators (TPEN, EDTA, HEDTA, EGTA, and NDAP) with 0.05–0.95 mM Zn(II) (metal buffers) over a period of 10 h [14,50]. Metal buffer sets were prepared in 20 mM Tris-HCl with 100 mM NaCl, 200 μM TCEP, and pH. 7.4. The equilibrated samples were measured in a 2 mm quartz cuvette at a fixed wavelength of ~220 nm, which slightly differed for each peptide in order to obtain the highest possible dynamic range of ellipticity changes. The samples were measured in kinetic mode in order to obtain 50 independent measurements, which were subsequently averaged to final values. The amount of Zn(II) transferred from the metal buffer component to a particular zinc finger peptide was considered while recalculating final free Zn(II) values. All -log[Zn(II)]_free_ (pZn) calculations were performed based on previously established dissociation constants of Zn(II) complexes with chelators using HySS software [51,52]. All experimental points recorded for each ZF were fitted to Hill’s equation, Equation (4) [12,53].
(4)$$\Theta =\Theta_{\min}\;\left(\frac{x^{n}}{x^{n}+{\left[\mathrm{Zn}{\left(\mathrm{II}\right)}_{0.5}\right]}^{n}}\right)+\Theta_{\max}\;\left(\frac{{\left[\mathrm{Zn}2{+}_{0.5}\right]}^{n}}{x^{n}+{\left[\mathrm{Zn}2{+}_{0.5}\right]}^{n}}\right)$$
where Θ_min_ is minimum ellipticity; Θ_max_ is maximum ellipticity; *n* is cooperativity index (Hill’s coefficient); x is a free Zn^2+^ concentration at a specific experimental point; and [Zn(II)_0.5_] is free Zn(II) concentration at the half-point saturation of the ZnL complex [50].

The obtained concentrations of free Zn(II), which correspond to the half-point ZnL complex saturation [Zn(II)_0.5_], were subsequently used to calculate the apparent dissociation constants *K*_d_ of the ZnL complexes based on the following Equation (5):
(5)$$\mathrm{Zn}(\mathrm{II})\;+\;L\;⇌\;\mathrm{ZnL}\;K_{d}=\frac{\left[\mathrm{Zn}\left(\mathrm{II}\right)\right]\;[L]}{\left[\mathrm{ZnL}\right]}$$

### 3.8. Isothermal Titration Calorimetry (ITC)

The ITC titrations were performed using a nano-ITC calorimeter (TA Waters), at 25 °C, in a measuring cell of 1 mL. All titrations were performed in 50 mM HEPES buffer (*I* = 0.1 M for NaCl) at pH 7.4 under an argon atmosphere. The concentration of peptide (analyte) in the measuring cell was within 30–35 µM and the Zn(II) (titrant) concentration in the burette was 0.350 mM. For these parameters, Zn(OH)_2_ precipitation was not observed. After equilibration and stabilization of the baseline, titration was carried out by injecting 6.82 µL of the titrant solution at 400 s intervals with stirring at 200 RPM. Control experiments to determine the heat of titrant dilution were performed using identical injections in the absence of Zn(II). The net reaction heat was obtained by subtracting the heat of dilution from the corresponding total heat of reaction. The titration data were analysed using NanoAnalyze (version 3.3.0) and Origin software (version 8.1, Northampton, MA, USA) and were fitted to a sequential binding model to account for the formation of ZnZF during the course of titration. All of the experimental enthalpy (Δ*H*_ITC_) values presented in Table 2 (main text) were used to calculate other thermodynamic parameters of the Zn(II) binding process. The process of Zn(II)-ZF complex formation is associated with deprotonation of the cysteine thiols. In this case, the experimental Δ*H*_ITC_ value is the sum of enthalpy changes induced during complexation (Δ*H*°) and during buffer protonation (*n*_H_Δ*H*°_buff_) according to Equation (6) [48]:(6)$$\Delta H_{\mathrm{ITC}}=\Delta H^{{}^{\circ}}+n_{H}\Delta H_{\mathrm{buff}}^{{}^{\circ}}$$
where $n_{H}$ value is associated with numbers of protons released during cysteine thiols’ deprotonation associated with Zn^2+^ binding and is equal to the number of protons associated with two thiol groups from the peptide at pH 7.4. The *n*_H_ value can be calculated using $pK_{a_{1}}^{\mathrm{SH}}$ and $pK_{a_{2}}^{\mathrm{SH}}$ values of each cysteine from each peptide [48]. To obtain these values, spectrophotometric pH-titration of Cys-containing peptides in the UV range was performed (Appendix A). The results from pH-titration were fitted to two binding event equations to obtain $pK_{a_{1}}^{\mathrm{SH}}$ and $pK_{a_{2}}^{\mathrm{SH}}$ (Equation (3)); Δ*H*°_buff_ is the heat of buffer protonation, which is specific to each buffer (for HEPES it is 5.02 kcal/mol) [54]; Δ*H*° is the sum of the enthalpy of metal ion complexation (Δ*H*°_Zn-pep_) and the enthalpy associated with the thiol deprotonation process (Δ*H*°_CysH_ = 8.5 kcal/mol) [31,48] multiplied by the number of protons associated with thiol deprotonation (*n*_H_), giving Equation (7):(7)$$\Delta H^{{}^{\circ}}=\Delta H_{\mathrm{Zn}-\mathrm{pep}}^{{}^{\circ}}+n_{H}\Delta H_{\mathrm{CysH}}^{{}^{\circ}}$$

The thermodynamics of metal ions binding to ZFs contribute to their structure stabilization; thus, it involves contributions from both metal coordination and interactions among the peptide residues. The Δ*H*°_Zn-pep_ quantifies the enthalpy of Zn(II) binding to peptides with deprotonated Cys residues, and includes both the Zn-ligand bond enthalpies (relative to the Zn-OH_2_ bond enthalpy) and the remaining enthalpy associated with the peptide folding (Δ*H*°_folding_). Since the Zn(II) coordination in examined ZFs is known, its contributions can be estimated from the enthalpy of Zn-N (Δ*H*°_Zn-N_) and Zn-S (Δ*H*°_Zn-S_) bonds. The Δ*H*° for Zn(II) binding to a His imidazole and a Cys thiolate are both approximately −5.0 kcal/mol [31,48]. Therefore, Δ*H*°_Zn-pep_ may be presented by Equation (8):(8)$$\Delta H_{\mathrm{Zn}-\mathrm{pep}}^{{}^{\circ}}=\left(4-n_{\mathrm{His}}\right)\Delta H_{\mathrm{Zn}-S}^{{}^{\circ}}+n_{\mathrm{His}}\Delta H_{\mathrm{Zn}-N}^{{}^{\circ}}+\Delta H_{\mathrm{folding}}^{{}^{\circ}}$$
where $n_{\mathrm{His}}$ is the number of His ligands present in ZF coordination sphere. In the next step of thermodynamic analysis, the entropic component (−T∆S°) was determined in accordance with the Gibbs free energy Equation (9):(9)$$\Delta G^{{}^{\circ}}=\Delta H^{{}^{\circ}}-T\Delta S^{{}^{\circ}}=-\mathrm{RTln}K_{b}$$
where R = 8.314 J mol^−1^·K^−1^, T = 298.15 K, and *K*_b_ is a value of binding (formation) constant for the ZnZF complex (Appendix A, Table 1). The *K*_b_ values were used to calculate the Gibbs free energy component (∆*G*°).

## 4. Conclusions

Our study points out the important role of non-conserved amino acids in ZFs stability adjustment. As proved by computational and experimental methods, we showed that additional h-bond interactions are formed between non-conserved residues derived from the α-helical part of ZFs. Moreover, we showed that Zn(II) binding to CCHH ZFs can also be entropically driven. This preference does not correlate with the Zn(II) coordination site or the extent of the secondary structure. Still, it is strictly related to a reservoir of interactions within the protein core (mediated by “second shell” side chains) which may loosen or tighten the structure. These can influence backbone configurational entropy, destabilizing Zn(II)-ZF complex formations. The overall stability of CCHH ZFs is modulated by decreasing or increasing the number of non-covalent interactions between specific non-conserved residues, found both in β-sheet or α-helix regions, while maintaining highly ordered ββα folds. Having this in mind, we showed that conserved residues and non-conserved residues can alter ZF stability, making ZF scaffold a highly flexible target, especially for biochemical engineering applications and toxicological studies. The presented results help us understand the rules of ZFs saturation under cellular conditions, indicating the dynamics of ZF metalloproteome.

## Figures and Tables

**Figure 2 ijms-23-14602-f002:**
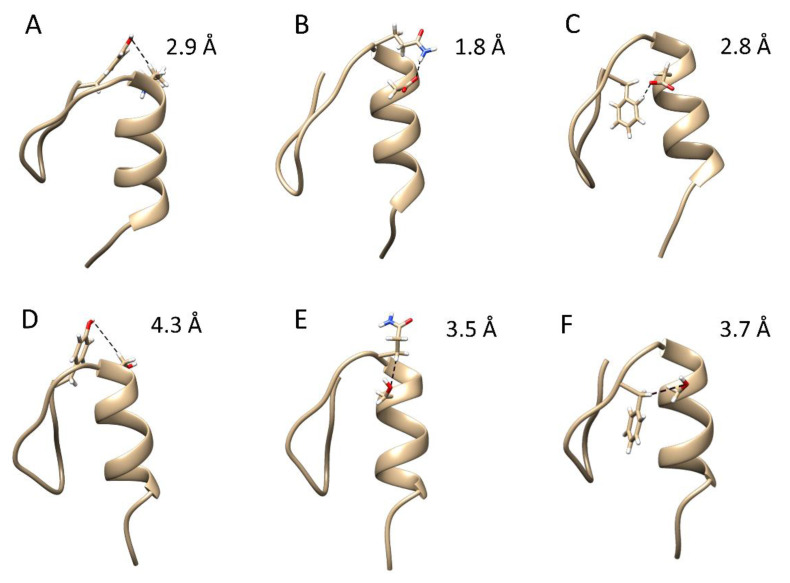
Intermolecular interactions obtained for CP1-1991. These were h-bond Asp15 (S, OD2)-Gln12 (S, HE21) (**A**), h-bond Asp15 (S, OD1)-Phe10 (S, HD2) (**B**), and h-bond Tyr1 (S,OH)-Lys13 (S,HB2) (**C**). As comparison, we have shown the lack of corresponding h-bonds in CP1-2015 (**D**–**F**). Note Ser15 and Ser13 replace Asp15 and Lys13 in CP-2015. Zn(II) and Zn(II)-binding residues were omitted in the graphics for clarity.

**Figure 3 ijms-23-14602-f003:**
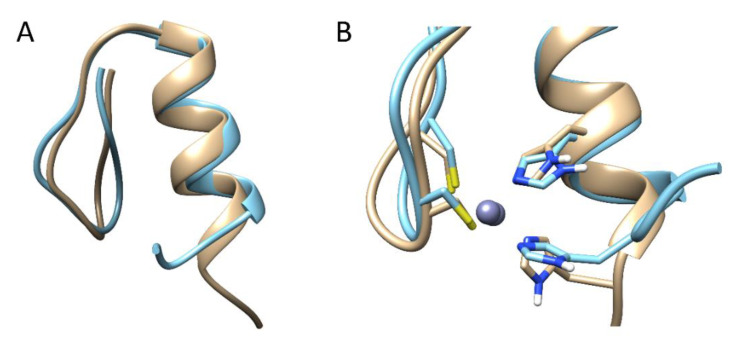
CP1-1991 (brown) and CP1-2015 (cyan) representative structures from cluster analysis performed over structures extracted from minimum energies from the FES. In (**A**), the Zn(II) ion was omitted for clarity and the full peptide sequence was shown, while (**B**) shows a zoomed region of the Zn(II) binding site.

**Figure 4 ijms-23-14602-f004:**
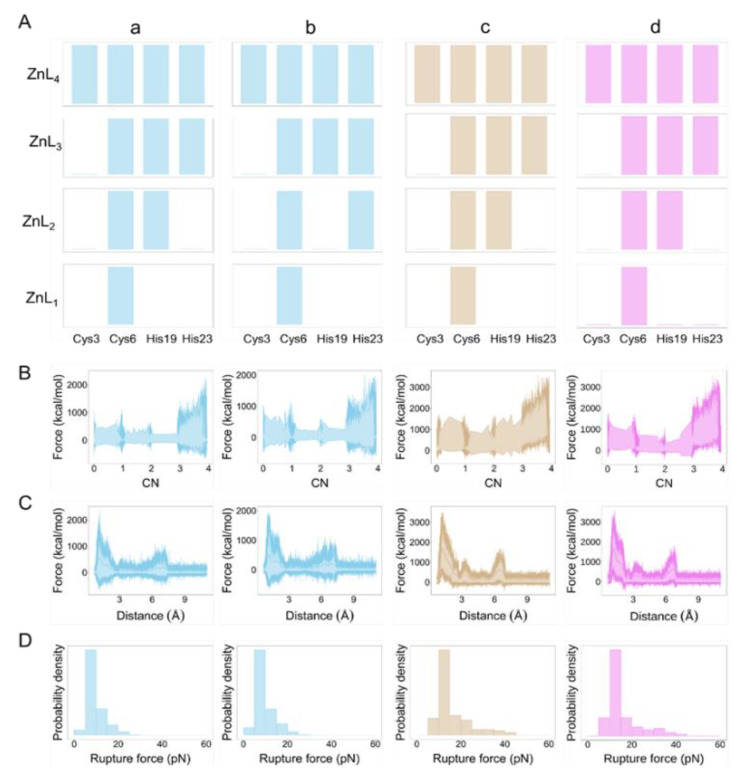
Steered MD simulations for the ZFs CP1-2015 and CP1-1991. (**A**) Stepwise ZnL dissociation for the CP1-1991 (**a**), CP-2015 pathway I (**b**), pathway II (**c**), and the CP1-1991α (**d**). The bar represents the case when Zn(II) is bound to a particular ligand (L). (**B**) Force rupture as a function of the contact number (CN). CN was defined as the number of ligands bound to the Zn(II) ion. (**C**) Force–extension curves derived from the SMD simulations. (**D**) Rupture force histograms for each pathway.

**Figure 5 ijms-23-14602-f005:**
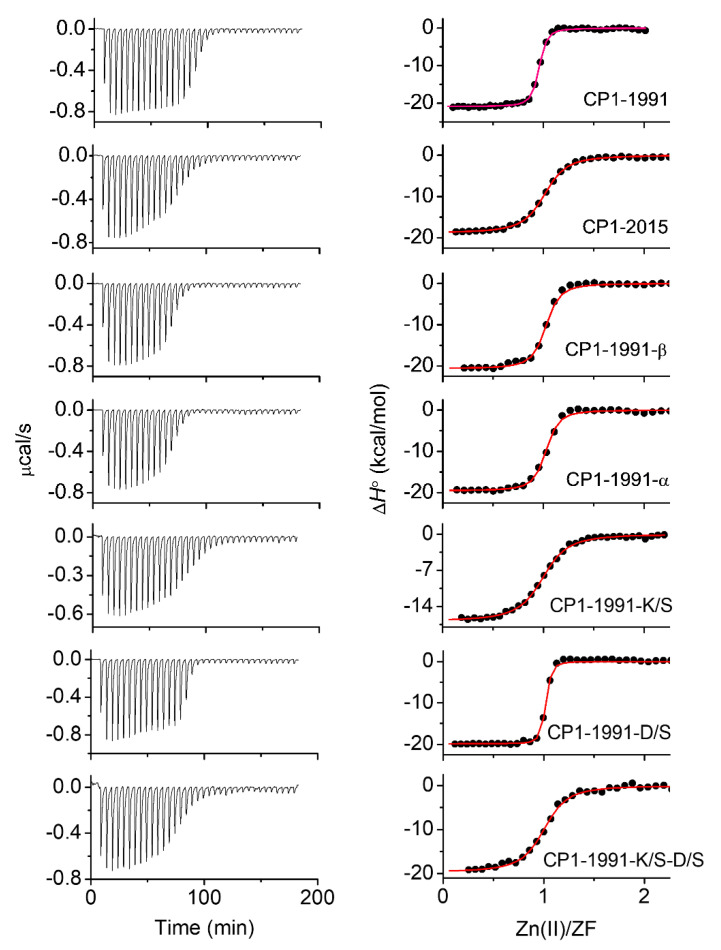
ITC binding isotherms (**left panel**) and best fits (**right panel**) for Zn(II) titrations into examined ZFs with multiple mutations: CP1-1991 (n = 0.99), CP1-2015 (n = 0.99), CP1-1991-α (n = 0.82), CP1-1991-β (n = 0.99) and point mutations: CP1-1991-K/S (n = 0.98), CP1-1991-D/S (n = 0.98), CP1-1991-K/S-D/S (n = 0.98) in 50 mM HEPES *I* = 0.1 M (from NaCl), pH 7.4 at 25 °C.

**Figure 6 ijms-23-14602-f006:**
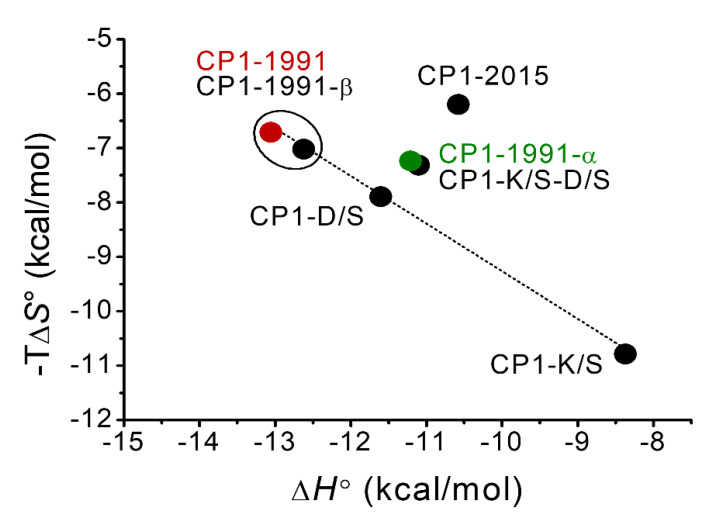
A plot of the entropy vs. enthalpy of zinc complexation by the ZF peptides at 25 °C and pH 7.4 in 50 mM HEPES (100 mM NaCl).

**Table 1 ijms-23-14602-t001:** Dissociation constant values of Zn(II) complexes with ZF peptides were determined spectropolarimetrically in the competition with chelators in 20 mM Tris buffer pH 7.4 (0.1 M NaCl) [14]. ∆p*K*_d_ values are differences of -log*K*_d_ of CP1-1991 and particular ZF.

ZF Peptide	−log*K*_d_^7.4^	*K*_d_^7.4^ (fM)	∆(−log*K*_d_^7.4^)
CP1-1991	14.49 ± 0.05	3.2 ± 0.4	0
CP1-2015	12.30 ± 0.02	501 ± 23	2.19
CP1-1991-β	14.40 ± 0.04	4.0 ± 0.4	0.09
CP1-1991-α	13.50 ± 0.05	32 ± 3	0.99
CP1-1991-K/S	14.04 ± 0.05	9.1 ± 0.9	0.45
CP1-1991-D/S	14.29 ± 0.04	5.1 ± 0.4	0.20
CP1-1991-K/S-D/S	13.52 ± 0.04	30 ± 3	0.97

**Table 2 ijms-23-14602-t002:** Enthalpy and entropy parameters of Zn(II)-to-ZF peptide binding derived from ITC measurements in 50 mM HEPES buffer pH 7.4 (100 mM NaCl). ΔΔ*G*° values are differences of Δ*G*° of particular ZF and Δ*G*° of CP1-1991.

ZF Peptide	Δ*G*°	ΔΔ*G*°	−TΔ*S*°	Δ*H*_ITC_	Δ*H*°	Δ*H*_CysH_	Δ*H*_Zn-pep_	Δ*H*_folding_	n_H_
	(kcal/mol)								
CP-1991	−19.77 ± 0.05	0	−6.71	−20.87 ± 0.19	−13.06	13.23	−26.29	−6.29	1.56
CP-2015	−16.78 ± 0.02	2.99	−6.20	−18.83 ± 0.39	−10.58	13.97	−24.44	−4.55	1.64
CP-1991-β	−19.64 ± 0.04	0.13	−7.02	−20.61 ± 0.40	−12.62	13.53	−26.15	−6.15	1.59
CP-1991-α	−18.42 ± 0.05	1.35	−7.31	−19.53 ± 0.43	−11.10	14.27	−25.37	−5.37	1.68
CP-1991-K/S	−19.15 ± 0.05	0.62	−10.78	−16.82 ± 0.20	−8.37	14.31	−22.68	−2.68	1.68
CP-1991-D/S	−19.49 ± 0.04	0.28	−7.89	−20.24 ± 0.35	−11.60	14.01	−25.62	−5.62	1.65
CP-1991-K/S-D/S	−18.44 ± 0.04	1.33	−7.23	−19.68 ± 0.47	−11.21	14.34	−25.55	−5.55	1.69

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
