# Peer review of "Non-Conserved Amino Acid Residues Modulate the Thermodynamics of Zn(II) Binding to Classical ββα Zinc Finger Domains"

_ijms, 2022, doi:10.3390/ijms232314602_

Round 1

Reviewer 1 Report

Reviewing report Manuscript ID: ijms-2041423

·         brief summary

This paper aimed to investigate how variation in the two consensus zinc finger peptide sequences impact the thermodynamics of zinc binding. This paper claimed that some non-conserved amino acid residues can modulate the thermodynamics of zinc binding

·         General concept comments

Nevertheless, if the authors indeed showed a difference in thermodynamics between both peptides, it is not clear that the residues they pointed out (notably Lys13 and Asp15), are the residues responsible for these differences in thermodynamics. First observation, the mean rupture forces and total work done do not seem significantly different between CP1-1995 and CP1-1995α, suggesting that the residues of the α-helix do not seem to be involved in the improved stability of CP1-1995. Second observation, even though individual mutation Lys13 did lead to a significant decrease in ΔH, this did not seem to be the case for the double Lys13 Asp15 mutant. If indeed h-bonding of both these residues was the major contributor to improved stability of ZF, the ΔΔH should be more significant. In conclusion, the authors did not support their claim with sound statistical analysis of their data, they should either prove their point with sound statistical analysis or reinterpret their data.

·         Specific comments

·         Line 59. Figure 1 legends says “Zn(II) binding (green)” but the residues are not colored in green in the figure. Change the legend and/or the figure.

·         Lines 68 and 69 : missing references.

·         Line 122. The figure should compare the H-bonds between both peptides, as it is not clear for me that the S13 could not make a H-bond with Tyr1 in the CP1-2015 peptide, similarly to what is observed with K13 and Tyr1.

·         Line 124. It is not explained how a h-bond in the N-terminal part of the α-helix effect stability at the other end of the helix. Why would residue 22 not rather be involved in this stabilization ?

·         Line 148. ZnL2 is missing after ZnL1 and before ZnL3.

·         Line 148. In Figure S3, A, B, and C are not explained, remove them maybe ? In ZnL3, we still see 4 ligands, hide one ligand.

·         Line 150. Table S1 should be Table S2.

·         Line 156. Table S1 should be Table S2. There was statistical analysis of the results, no Student test or explanation of what represent the +.

·         Line 157. Table S1 should be Table S2. Legend of Figure S4 is incomplete. There was statistical analysis of the results, no Student test or explanation of what represent the +. You cannot conclude that the total work done follows a similar trends.

·         Lines 171-173. As the mean rupture force are similar for CP1-1995 and CP1-1995a (Table S2), this suggests that the h-bond formation between non-conserved residues found in the C-terminus α-helix are not involved in the stabilization of CP1-1995, as these residues are not present in CP1-1995α. Your claim is not founded.

·         Lines 253-255. This claimed is not founded. Mutation of both Lys13 and Asp15 only lead to a very slight decrease in ΔH. Individual mutation of Lys13 do lead to a significant decrease in ΔH, but this seem to be compensated by the Asp15 mutant. This observation should be explained.

·         Lines 323-324. I do not see overwhelming evidence that the ZF scaffold is highly flexible. Please develop.

Reviewer 2 Report

This is a comprehensive experimental and computational investigation of factors that influence the binding of Zn to Zn-finger proteins.  The data and analysis make a substantial contribution to the understanding of these proteins,  and to their manipulation.

I recommend acceptance.

I note a few minor improvements.

line 16-17:  ‘medium affinity causes folds transient saturation’  this should be expressed more clearly

line 34:  CCHH not defined, for the general reader

line 235:  change ‘loose’  to ‘loss’

Author Response

Reviewer #2

This is a comprehensive experimental and computational investigation of factors that influence the binding of Zn to Zn-finger proteins.  The data and analysis make a substantial contribution to the understanding of these proteins,  and to their manipulation.

I recommend acceptance.

I note a few minor improvements.

1) line 16-17:  ‘medium affinity causes folds transient saturation’  this should be expressed more clearly

Answer: This sentence has been modified.

2) line 34:  CCHH not defined, for the general reader.

Answer: The reviewer is right. We changed this sentence by adding (Cys2His2 or CCHH) after the phrase “by pair of cysteine and histidine residues.” Moreover, we have defined CCHH in the Abbreviation section.

3) line 235:  change ‘loose’  to ‘loss’.

Answer: The word has been replaced.

Reviewer 3 Report

According to the author’s in vitro and in silico experiments, non-conserved residues can change metal-coupled folding mechanisms and overall ZF stability. In this manuscript, the authors can  able to demonstrate how sequential composition affects heterogeneity throughout ZFs stability and describe which non-conserved residues are crucial in stability alteration without significantly affecting ZF structure. Furthermore, the authors demonstrate that entropic forces can also be used to induce Zn(II) binding to CCHH ZFs. This choice is not related to the size of the secondary structure or the Zn(II) coordination site. However, it is purely connected to a pool of interactions that can either loosen or tighten the structure within the protein core.

I personally believe that this paper will provide better understanding ZF-DNA interaction and have implications in protein design. The paper is well written, methods are clearly described, and results are well discussed. The data presented supports the conclusion which is much enough to make a strong statement as a title (Non-conserved amino acid residues modulate the thermodynamics of Zn(II) binding to classical zinc fingerdomains. I recommend the paper for publication.

Author Response

According to the author’s in vitro and in silico experiments, non-conserved residues can change metal-coupled folding mechanisms and overall ZF stability. In this manuscript, the authors can  able to demonstrate how sequential composition affects heterogeneity throughout ZFs stability and describe which non-conserved residues are crucial in stability alteration without significantly affecting ZF structure. Furthermore, the authors demonstrate that entropic forces can also be used to induce Zn(II) binding to CCHH ZFs. This choice is not related to the size of the secondary structure or the Zn(II) coordination site. However, it is purely connected to a pool of interactions that can either loosen or tighten the structure within the protein core.

I personally believe that this paper will provide better understanding ZF-DNA interaction and have implications in protein design. The paper is well written, methods are clearly described, and results are well discussed. The data presented supports the conclusion which is much enough to make a strong statement as a title (Non-conserved amino acid residues modulate the thermodynamics of Zn(II) binding to classical zinc finger domains. I recommend the paper for publication.

Answer: We thank the reviewer for the positive reception of our work. Please note that we have improved the text according to Reviewer #1 and #2 suggestions.